# Characterization of *bla*_NDM-5_-and *bla*_CTX-M-199_-Producing ST167 *Escherichia coli* Isolated from Shared Bikes

**DOI:** 10.3390/antibiotics11081030

**Published:** 2022-07-30

**Authors:** Qiyan Chen, Zhiyu Zou, Chang Cai, Hui Li, Yang Wang, Lei Lei, Bing Shao

**Affiliations:** 1Beijing Key Laboratory of Detection Technology for Animal-Derived Food Safety, College of Veterinary Medicine, China Agricultural University, Beijing 100193, China; qiyanchen@cau.edu.cn (Q.C.); zouzhiyu@cau.edu.cn (Z.Z.); wangyang@cau.edu.cn (Y.W.); 2Beijing Key Laboratory of Diagnostic and Traceability Technologies for Food Poisoning, Beijing Center for Disease Prevention and Control, Beijing 100013, China; lihui@bjcdc.org; 3College of Arts, Business, Law and Social Sciences, Murdoch University, Perth, WA 6150, Australia; c.cai@murdoch.edu.au; 4Key Laboratory of Applied Technology on Green-Eco-Healthy Animal Husbandry of Zhejiang Province, Provincial Engineering Research Center for Animal Health Diagnostics & Advanced Technology, Zhejiang International Science and Technology Cooperation Base for Veterinary Medicine and Health Management, China Australia Joint Laboratory for Animal Health Big Data Analytics, College of Animal Science and Technology & College of Veterinary Medicine, Zhejiang A&F University, Hangzhou 311300, China

**Keywords:** shared bikes, NDM-5, CTX-M-199, ST167, whole genome analysis

## Abstract

Shared bikes as a public transport provide convenience for short-distance travel. Whilst they also act as a potential vector for antimicrobial resistant (AR) bacteria and antimicrobial resistance genes (ARGs). However, the understanding of the whole genome sequence of AR strains and ARGs-carrying plasmids collected from shared bikes is still lacking. Here, we used the HiSeq platform to sequence and analyze 24 *Escherichia coli* isolated from shared bikes around Metro Stations in Beijing. The isolates from shared bikes showed 14 STs and various genotypes. Two *bla*_NDM-5_ and *bla*_CTX-M-199_-producing ST167 *E. coli* have 16 resistance genes, four plasmid types and show >95% of similarities in core genomes compared with the ST167 *E. coli* strains from different origins. The *bla*_NDM-5_- or *bla*_CTX-M-199_-carrying plasmids sequencing by Nanopore were compared to plasmids with *bla*_NDM-5_- or *bla*_CTX-M-199_ originated from humans and animals. These two ST167 *E. coli* show high similarities in core genomes and the plasmid profiles with strains from hospital inpatients and farm animals. Our study indicated that ST167 *E. coli* is retained in diverse environments and carried with various plasmids. The analysis of strains such as ST167 can provide useful information for preventing or controlling the spread of AR bacteria between animals, humans and environments.

## 1. Introduction

Shared bikes as a public transport provide more choices and convenience for people’s travel. They also act as the last-mile connection between means of transport such as light rail stations or bus stops and people’s destinations such as home or the office. Some studies suggest that public transportation such as buses, subways, and taxis can act as a transmission media for bacteria or viruses [1,2], which could cause public health emergencies. Meanwhile, microorganisms on the surface of public transport arouses concern due to the severity of antimicrobial resistance worldwide [3,4,5]. Previous studies indicated that antimicrobial resistant (AR) Enterobacteriaceae, *Staphylococcus* spp. and *Enterococcus* spp. were already isolated from shared bikes [6,7,8,9]. Additionally, various bacteria with antimicrobial resistance genes (ARGs) were found in buses, subways, and aircrafts [1,2,3,4].

Several studies showed that both Gram-positive bacteria and Gram-negative bacteria could be isolated from shared bikes. Among them, *Staphylococci* and *Enterococci* were widely distributed in shared bikes around schools, hospitals, metro stations, and from riders, with detection rates of 2.3–12.9% and 0.08–5.5%, respectively [6]. The multiple resistant *Staphylococci* showed diversity in *SCCmec* and sequence type (ST) [8]. Meanwhile, the prevalence of Enterobacteriaceae in shared bikes was 19.7%, which suggested that hospitals might increase the risk of AR Enterobacteriaceae based on the distance from the hospital to the subway station [9]. Wu et al. reported that *Bacillus* was the most abundant bacteria in the shared bicycle bacteria community, and the drug-resistant bacteria in the shared bicycle bacterial community of metro stations, shopping malls, and hospitals showed no significant differences [7].

In recent years, the increasing reports of carbapenem resistance genes have increased the pressure on effective bacterial treatment. ST167 *E. coli* was often reported to carry carbapenem resistance genes such as *bla*_KPC-3_, *bla*_NDM-5_, and *bla*_NDM-1_ and was found in various species such as ducks, cattle, and mussels [10,11,12,13,14,15]. A study on hospitalized neonatal sepsis showed that *E. coli* (34.01%) was one of the main pathogens of neonatal bacteremia, and ST167 was the most prevalent ST [16]. More importantly, ST167 has been reported to spread between companion animals and their owners [12]. The spread of ST167 clones between countries has also been reported [17]. Although characterization of bacteria from shared bikes has attracted widespread attention in recent years, current studies have mainly focused on the prevalence and the phenotypes of strain descriptions in public transportation or the features of isolates themselves. To the best of our knowledge, the whole genome analysis with strains from different locations or biological sources and comparisons of their plasmid profiles are still lacking. *E. coli* is an important representative of Enterobacteriaceae, which can carry a variety of ARGs and has significance for public health safety. Herein, we used the *E. coli* isolates from the shared bikes to investigate the similarities and differences between strains from the shared bikes and other sources to find the relationship of the whole genome sequencing between the *E. coli* isolates from environmental and clinical samples.

## 2. Results and Discussion

### 2.1. E. coli Isolates from Shared Bikes

We identified 14 STs among all 24 *E. coli* isolates from shared bikes (14 from Metro Station nearing secondary/tertiary hospitals and ten from non-hospital stations, Appendix A), and the ST10 clonal complex (n = 7) were the dominant clonal complex (Figure 1). There is no dominant ST or clonal complex related to hospitals, although ST10, ST48, and ST167 found in this study were the most prevalent STs in hospitals [18,19,20].

The phylogenetic tree analysis showed that the 24 *E. coli* strains from shared bikes had different profiles. The number of ARGs in each of the strains ranged from one to sixteen, and plasmid types ranged from zero to five (Appendix A). The resistance phenotypes showed that some strains (such as 770, 776) which have a higher number of resistance genes exhibited more resistance to antimicrobial agents than other strains. However, the number of strains exhibiting resistance phenotype mismatch the number of ARGs. Some strains showed high similarity in one small clade, for instance, 26, 25, 31 and 769, 780. Almost all AR strains have resistance genes of aminoglycosides, quinolones, sulfonamides, tetracyclines and beta-lactams. Despite most strains (66.7%) from hospital-related stations have resistance gene to different kinds of antimicrobial agents, there is no significant difference between multidrug resistance (MDR) *E. coli* from hospital-related stations and non-hospital stations (*p* > 0.05). The two strains (770 and 776) collected, respectively, from hospital-related stations and non-hospital stations carried the maximum number of resistance genes and plasmid types of all strains and showed >95% similarities in core genomes with the same sequence type ST167 (Figure 2).

Furthermore, these two strains carried *bla*_NDM-5_ and *bla*_CTX-M-199_, and another 14 resistance genes including aminoglycoside resistance genes *aadA2*, *aadA5*, *aph(3″)-Ib*, *aph(6)-Id*, *rmtB* beta-lactam resistance genes *bla*_EC-15_ and *bla*_TEM-1_, phenicol resistance gene *floR*, macrolide resistance gene *mph*(A), tetracycline resistance gene *tet*(A)*,* sulfonamide resistance genes *sulI*, *sulII*, trimethoprim resistant genes *dfrA12* and *dfrA17*. In addition, the comparison of virulence factors between hospital-related and non-hospital stations showed no significant difference (*p* > 0.05). ST167 is one of the epidemic STs in *E. coli* that carried ARGs, especially β-lactamase genes [21]. Previous studies indicated that ARGs-carrying ST167 *E. coli* were isolated from humans, food animals, companion animals and environments [10,15,22,23]. Until now, ST167 *E. coli* were found in countries and districts across five continents, such as China, Tunisia, Switzerland, Italy, Finland, Canada, Brazil, and Tanzania [22,23,24,25,26,27,28]. The ST167 *E. coli* carrying the *bla*_NDM_ gene were previously identified in hospitals, livestock farms, poultry farms, and the environment [10,15,23,25,29]. Growing evidence indicated that the public environment is of increasing concern as a reservoir for the transmission of MDR bacteria and genes. However, unlike strains from farm environments, the AR bacteria strains from public transportation mean that they can be transferred between individual populations due to personnel movement.

### 2.2. Comparison of Core Genome with ST167 E. coli from Different Origins

Due to the high prevalence of ST167 *E. coli* in the world, we would like to compare the profiles of ST167 *E. coli* from shared bikes and from other origins (Appendix A). A total of 404 ST167 *E. coli* from the NCBI database were selected for comparative analysis with two *E. coli* from shared bikes. These strains were collected from human (n = 370), food animals (n = 11), companion animals (n = 15), environment (n = 8) samples (Figure 3) from 35 countries or districts (Appendix A). More than half of strains carried *bla*_NDM_ (n = 288) and *bla*_CTX-M_ gene (n = 272). *bla*_NDM-5_ were the most prevalent NDM type (n = 254) but *bla*_CTX-M-199_ were found on only one strain. The phylogenetic tree indicated that all ST167 *E. coli* strains exhibited various characterizations in the core genome and have 21~11,206 single nucleotide polymorphisms (SNPs) compared to strains from shared bikes. The two ST167 *E. coli* from shared bikes show high similarity (SNPs < 50) with 33 strains (Pink color range in the Figure 3) from samples of human (n = 18), dogs (n = 12), cats (n = 1), chicken (n = 1) as well as environment (n = 1). The human samples were identified from Bangladesh (n = 1), the United Kingdom (n = 7), China (n = 6), Switzerland (n = 1), Italy (n = 2) and the United States (n = 1). The dog strains were originated from Switzerland (n = 2) and the United States (n = 10). Other strains were collected from a cat in Italy, a chicken in China and an environmental source from the United States. All strains were collected from 2015 to 2021, while 32 of these strains carried *bla*_NDM-5_.

### 2.3. Comparison of Plasmid Profiles with ST167 E. coli from Different Origins

Illumina and Nanopore sequencing of *bla*_NDM-5_ or *bla*_CTX-M-199_-carrying isolates indicated that *bla*_NDM-5_ and *bla*_CTX-M-199_ were located on a ~98.5 kb IncFII plasmid and a ~113 kb IncFII plasmid, respectively. From the NCBI database, we downloaded nine plasmids that have the highest coverage and identities in sequences with *bla*_NDM-5_- or *bla*_CTX-M-199_-carrying plasmid of shared bikes (Appendix A). Nine *bla*_NDM-5_-carrying plasmids belong to strains from patients in China (n = 3), Japan (n = 1), Tanzania (n = 1), Myanmar (n = 2) and Switzerland (n = 1), and from chicken meat in Laos (n = 1). Plasmid pNDM-EC16-50 in one *E. coli* strain from China showed >90% coverage and highest identifies with the *bla*_NDM-5_-carrying plasmid of shared bikes (Figure 4a). Nine *bla*_CTX-M-199_-carrying plasmids belong to strains from human (n = 3), chicken (n = 1), goose (n = 1) in China, humans in the United States (n = 1), Japan (n = 1), and Lebanon (n = 1), and water samples from India (n = 1). The nucleotide sequence of the *bla*_CTX-M-199_-carrying plasmid of shared bikes displayed the highest similarity with *E. coli* strain L100 plasmid pL100-3 and *E. coli* plasmid J-8 plasmid pCTX from goose and chicken in China (Figure 4b). According to the information of NCBI, we download the isolates that carried these similar plasmids (*bla*_NDM-5_ or *bla*_CTX-M-199_) and identified the ST of these isolates (except nine plasmids without the whole genome of isolates upload). The results showed that three *bla*_NDM-5_ plasmids were from ST167 *E. coli* of human origin, the other three plasmids were from non-ST167 *E. coli,* and three *bla*_CTX-M-199_ plasmids were from ST10 *E. coli* of human origin, ST148, ST156 *E. coli* of food animal origin. The results of the plasmid profiles comparison indicated that maybe some bacteria carrying plasmids with ARGs from patients and farm animals are possible to persist in the environment and further plasmid conjugative transfer to the bacteria of environments. Moreover, ST167 can acquire plasmids easily from other STs, which makes plasmids with ARGs commonly available.

The plasmid analysis of ST167 *E. coli* from shared bikes showed that ST167 might be an important strain for plasmid-borne ARGs across different origins. Furthermore, combined with the results of the phylogenetic tree, the strains which have good environmental adaptability can increase the possibility of plasmid transfer between different bacteria which enhances the dissemination of ARGs among animals, humans and the environment, and threaten public health. Researchers are also concerned about the AR bacteria and gene transmission via the environment [30,31,32]. Furthermore, these strains increased the difficulty of AR control. However, not only can ST167 act as the vector but also some other prevalent strains can play the same role as ST167, so the control of prevalent strains requires substantial concern.

We found a high similarity of strains from the shared bikes and other origins, which means that some well-adapted isolates can persist in different environments. Many studies proved that AR bacteria isolated from the same environment are convergence in molecular profile because the environment has prevalent AR bacteria and genes [33]. However, now some prevalent STs such as ST131, ST167 and ST 10 *E. coli* which have good environmental adaptability with ARGs can be a potential reservoir of ARGs in the environment [13]. Moreover, ST 167 *E. coli* carried important resistant genes, such as *bla*_NDM_, *mcr-1*, *bla*_CTX-M_ [33], which are stable in the environment and pose a threat to public health. The flow of population further accelerates ARGs spread to diverse environments or different species, which adds pressure to control antimicrobial resistance. Therefore, the dominant host of ARGs like ST 167 in the environment should be concerned and focused on.

## 3. Materials and Methods

### 3.1. Bacterial Isolates, Whole Genome Sequencing

A total of 444 Enterobacteriaceae were isolated from shared bikes in the previous study and *E. coli* was the species that exhibited more drug resistance than others [9]. Therefore, *E. coli* was chosen for further analysis. A total of 28 *E. coli* strains were isolated from samples in the previous study, excluding 4 from stations outside the fifth Ring Road of Beijing; finally, 24 *E. coli* isolates were collected. Genomic DNA was extracted using a HiPure Bacterial DNA Kit. DNA libraries were prepared and sequenced with HiSeq PE150. Two *bla*_NDM-5_ and *bla*_CTX-M-199_-producing *E. coli* were sequenced with Nanopore to obtain the complete plasmids. The sequences were assembled by SPAdes and Unicycler.

### 3.2. Assembled Data of ST167 E. coli from Different Sources

We searched all *E. coli* available in the NCBI database which were collected from January 2014 to December 2021 and downloaded those. We only selected assembled data of whole genome sequencing. Furthermore, we reorganized the detailed information related to the assembled data we downloaded and excluded strains without information on the host. All genomes were confirmed the ST using MLST. Additionally, only *E. coli* with ST167 were chosen for the following analysis.

### 3.3. Genomic Analysis of Sequenced and Collected E. coli Strains

ARGs and plasmid incompatibility groups were determined using the database (resfinder, plasmidfinder) from the Center for Epidemiology (http://www.genomicepidemiology.org/, accessed on 12 February 2022). According to the mechanism of resistance classified ARGs in different antibiotic classes. MLST was confirmed using MLST in the Center for Epidemiology and database from public databases for molecular typing and microbial genome diversity (https://pubmlst.org/, accessed on 23 February 2022 ). The Sankey diagram of the ST clonal complex was performed using plug-in components of Excel named EasyShu. Virulence genes were identified using the VFDB database and virulencefinder of Center for Epidemiology. The criteria of different groups of virulence genes in accordance with VFDB. The tests were used for the comparison of the number of virulence factors from hospital-related Stations and non-hospital Stations. Core genomes were extracted using Snippy [34]. Core genome phylogenetic trees were constructed using Snippy and Fasttree [35]. Phylogenetic tree of the core genomes with ARGs, plasmid types, stations and phenotypes displayed with iTOL [36]. Genes in the plasmids were annotated using PATRIC and NCBI. The comparison of plasmid profiles was performed using BLAST and BRIG. The reference plasmid was annotated using the DNAplotter.

## 4. Conclusions

ST167 *E. coli* found on shared bikes showed high similarities with strains from patients and food-producing animals, and the plasmids also showed high identities with those from humans and animals in this study. These AR bacteria may originate from hospitals or farms. Vectors such as shared bikes may contribute to the dissemination of these AR bacteria in the environment. Furthermore, the persistence of these AR bacteria in the environment challenges the control of AR bacteria and ARGs. In the future, we need to take measures to assess the risk of AR bacteria in the environment and cut off transmission.

## Figures and Tables

**Figure 1 antibiotics-11-01030-f001:**
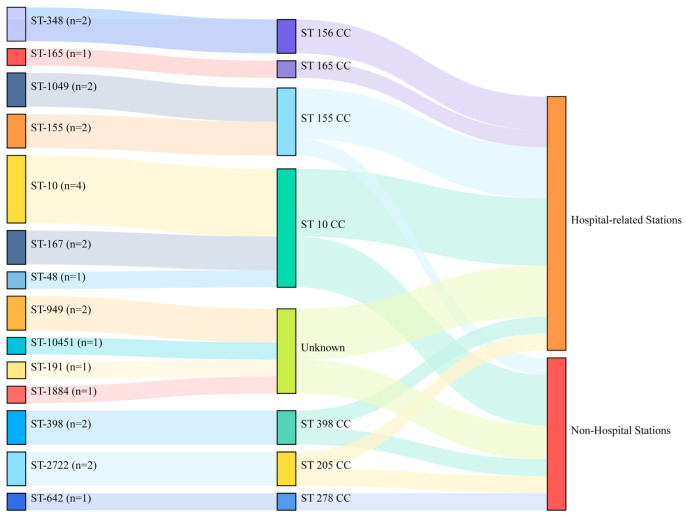
The distribution of STs from 24 *E. coli* in shared bikes. (CC: clonal complex, hospital-related stations represent Metro Stations nearing secondary/tertiary hospitals).

**Figure 2 antibiotics-11-01030-f002:**
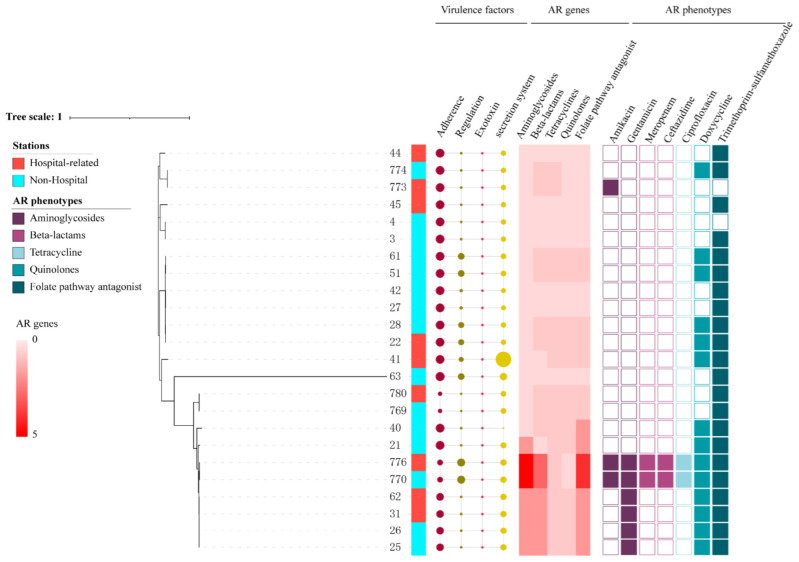
The phylogenetic tree of 24 *E. coli* from shared bikes. (The size of circles represents the number of virulence factors. The color of the heatmap indicated the number of resistance genes found in different antibiotic classes. Different colors were used to distinguish AR phenotypes of each antibiotic class).

**Figure 3 antibiotics-11-01030-f003:**
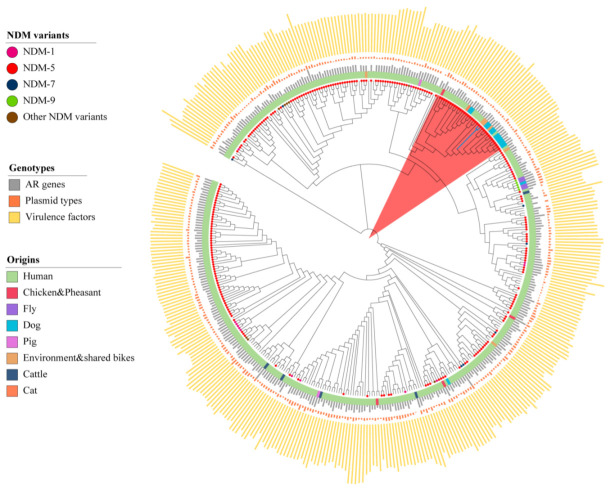
The core genome phylogenetic tree of ST167 *E. coli* from humans, animals and the environment. (The blue branches are the strains from shared bikes, the length of the bar represents the number of ARGs/Plasmid types/Virulence factors. The circles attached to the leaves represent the NDM variants).

**Figure 4 antibiotics-11-01030-f004:**
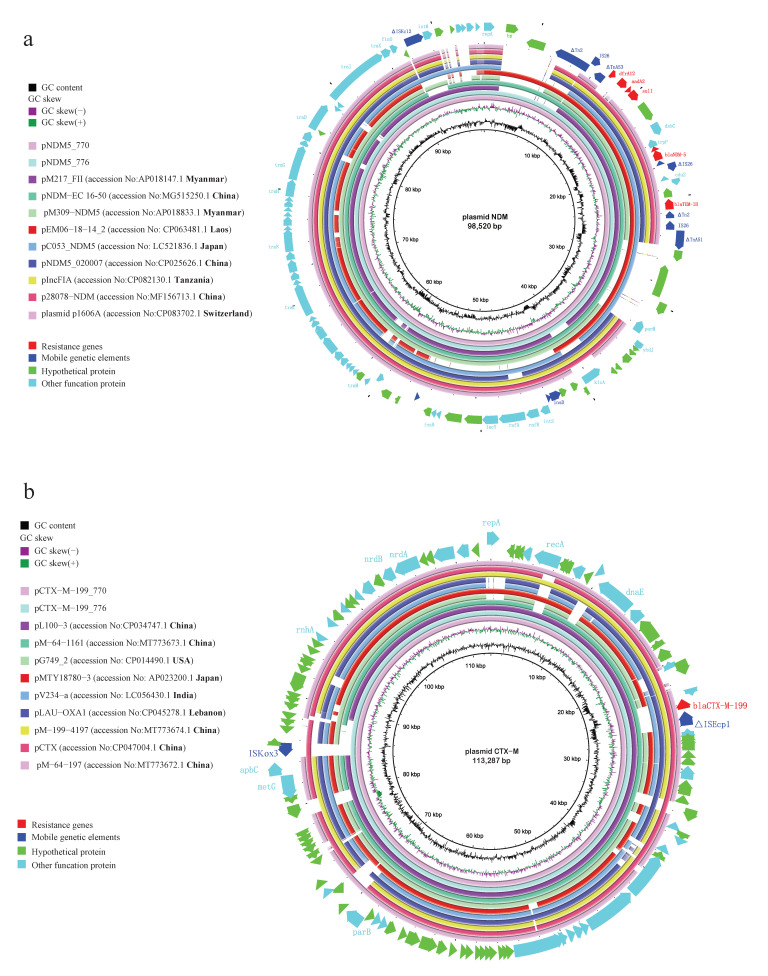
The plasmid profiles of (**a**) *bla*_NDM-5_ and (**b**) *bla*_CTX-M-199_. (The reference sequences were *bla*_NDM-5_- or *bla*_CTX-M-199_-carrying plasmid of shared bikes. The shade of circles represents the number of identities, the blank means sequences were not consistent with the reference).

## Data Availability

The data and material information used and analyzed in the current study are available from the corresponding author upon reasonable requests.

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
