# Peer review of "Characterization of *bla*_NDM-5_-and *bla*_CTX-M-199_-Producing ST167 *Escherichia coli* Isolated from Shared Bikes"

_antibiotics, 2022, doi:10.3390/antibiotics11081030_

Round 1
Reviewer 1 Report
This is a well thought study with implication on human health. It highlights potential spread of treatment resistant pathogens in our daily existence without being aware.
However, the conclusion is vague, devoid of the significance of the other sources used in the study and how they compare to shared bikes as vectors or contaminants. The conclusion should be improved and aligned to the title and study aim.
Is there any possibility of your finding informing or influencing public health policy?
Reviewer 2 Report
The research tackles an utmost important subject, that of AMR in the environment and tries to identify vehicles for the resistance plasmids as well as their distribution in some pathogenic bacteria (E. coli). Nevertheless, there are many gaps that hinder the precise understanding of the data obtained in the study.
In the Introduction, the purpose of paper should state that the subject is exclusively E. coli, not "Herein, we used isolates from the shared bikes to show the similarities and differences between strains from the shared bikes and other source to find the relationship between isolates from the environmental and from the clinical samples.", since it could be misleading for the reason of multiple species/strains' potential presence.
In the research design, it is not clear how many total E.coli isolates were identified in the previous research mentioned by the authors and why these 24 were selected.
In the Results and Discussion section the "2.3 Comparison of plasmid profiles with ST167 E. coli from different origins" could be more explicit, maybe a table to include all the information which in some places is confusing.
The English needs to be improved, since, in most places (i.e., the explanation of the results) it leads to confusion. The authors should make a better use of their results, stressing the importance and relevance of their findings.
Reviewer 3 Report
This manuscript characterised the drug resistance markers/genotypes of E.coli strains previously isolated from shared bikes. In light of the current AMR crisis and horizontal gene transfer of AMR genes, this article provides a critical understanding of how people share their commensal organisms with the community and what is prevalent among the population.
The genome sequencing of 24 E.coli strains isolated from shared bikes from different metro stations (hospital and non-hospital stations) identified AMR genes. In addition, they identified resistance genes to aminoglycosides, quinolones, sulfonamides, tetracyclines and beta-lactams. To investigate the transmission routes of MDR bacteria and genes, the author compared the core genome with MDR ST167 E.coli strain isolated from different origins, including human samples, food animals, companion animals and environment (sequences retrieved from NCBI). The author also compared the plasmid IncFII plasmid, IncFII 137 plasmid carrying blaNDM-5, blaCTX-M-199 resistant genes, respectively, from ST167 E. coli from different origins.
Comments:
-It would have been more interesting and informative if phenotypic resistance studies had been conducted to show whether the identified resistant genes can also exhibit resistance to antimicrobial agents.
The manuscript is somehow shorter in the context of introduction, results and discussion.
-Need to add figure legends and the figure can be slightly zoomed.
-The introduction is too short and needs more background information
-Methods section requires a bit of detail on each analysis.
Minor comment:
-More grammar and typos
Round 2
Reviewer 2 Report
Please find attached some minor comments to the revised version
